# The Efficacy of Daily Salmon Oil for Adult Type 2 Asthma: An Exploratory Randomized Double-Blind Trial

**DOI:** 10.3390/md23080328

**Published:** 2025-08-15

**Authors:** Katarina Mølsæter, Kjetil Roth, Tor Åge Myklebust, Erland Hermansen, Dave Singh, Crawford Currie, Dag Arne Lihaug Hoff

**Affiliations:** 1Department of Health Sciences, Faculty of Medicine and Health Sciences, Norwegian University of Science and Technology (NTNU), P.O. Box 1517, 6025 Ålesund, Norway; erland.hermansen45@gmail.com; 2Department for Research and Innovation, Møre og Romsdal Hospital Trust (HMR), P.O. Box 1600, 6026 Ålesund, Norway; tor.age.myklebust@helse-mr.no; 3Department of Clinical and Molecular Medicine, Faculty of Medicine and Health Sciences, Norwegian University of Science and Technology (NTNU), P.O. Box 8905, 7491 Trondheim, Norway; kjetil.roth@helse-mr.no; 4Department of Medicine, Ålesund Hospital, Møre og Romsdal Hospital Trust (HMR), P.O. Box 1600, 6026 Ålesund, Norway; 5Manchester University NHS Foundation Trust, Wythenshawe, Manchester M23 9L, UK; dsingh@meu.org.uk; 6Medicines Evaluation Unit, Langley Building, Southmoor Road, Wythenshawe, Manchester M23 9QZ, UK; 7Hofseth Biocare ASA, Keiser Wilhelms g. 24, 6003 Ålesund, Norway; cc@hofsethbiocare.no; 8Department of Clinical Studies, Ålesund Hospital, Møre og Romsdal Hospital Trust (HMR), P.O. Box 1600, 6026 Ålesund, Norway

**Keywords:** eosinophilic asthma, type 2-inflammation, pro-resolving lipid mediators, marine fish oil, adult, double-blind, randomized controlled trial

## Abstract

Asthma is a heterogeneous chronic respiratory condition with distinct inflammatory phenotypes, including type 2-driven eosinophilic asthma. This randomized, double-blind, placebo-controlled exploratory trial investigated the effects of OmeGO^®^, on respiratory outcomes in adults with type 2 asthma. Over a 20-week period, 66 participants received 6 g per day of either OmeGO^®^ (≥120 mg eicosapentaenoic acid (EPA), ≥180 mg docosahexaenoic acid (DHA)), or placebo. The key outcome was a composite score of moderate and severe exacerbation events. Asthma control was assessed using the Asthma Control Questionnaire-5-item (ACQ5) and the Global Initiative for Asthma (GINA) criteria. The median time to the first composite event was 37 days (95% CI 9–47) in the OmeGO group and 15 days (95% CI 12–33) in the placebo group (*p* = 0.347); 73% of the participants in the OmeGO experienced at least one exacerbation compared to 82% in the placebo group (*p* = 0.347). The weekly frequence of composite events was 0.36 per day in the OmeGO group and 0.32 in the placebo group (*p* = 0.777). Even though there are no differences in the exacerbation rates between groups, the time to first composite event should be further explored.

## 1. Introduction

Asthma is a chronic respiratory condition characterized by airway hyperresponsive-ness and inflammation presenting variable symptoms. These are typically wheezing, breathlessness, chest tightness, increased mucus production and coughing, with a fluctuation in both intensity and frequency [1]. Globally, asthma poses a substantial health burden, affecting more than 250 million people [2]. In Norway, adult asthma prevalence was recorded at 3.1% in 2018, with 5.4% of these cases identified as severe [3]. The American Thoracic Society (ATS) and the European Respiratory Society (ERS) recommend a stepwise treatment framework, as detailed by the Global Initiative for Asthma (GINA) [4]. This approach is adapted to the severity of asthma symptoms and the patient’s treatment response [2]. To minimize the risk of exacerbation, the recommended baseline regimen consists of inhaled corticosteroids (ICS) combined with formoterol, a long-acting beta-2 agonist (LABA). Potential treatment strategies include long-acting muscarinic antagonists (LAMAs), leukotriene receptor antagonists (LTRAs), and advanced biologic therapies targeting immunoglobulin E (IgE) or interleukin (IL) 5 pathways [2,4]. Difficult-to-treat asthma is defined as persistent symptoms and/or exacerbations despite optimal treatment according to GINA Step 4–5 guidelines [4,5,6]. Asthma control is commonly evaluated using standardized tools such as the GINA classification, the Asthma Control Test (ACT), and the Asthma Control Questionnaire (ACQ) [7].

The immune response (eosinophilic, allergic) in asthma is typically triggered by exposure to airborne allergens, and it is commonly classified into two main phenotypes: type 2 and non-type 2 [8]. The most common type of asthma is the type 2 phenotype characterized by increased levels of biomarkers like immunoglobulin E (IgE) and of eosinophils [1,6] driven by increased production of cytokines (interleukin IL4, IL5, and IL13) [1,8] Additionally, IL13 stimulates the production of nitric oxide, and the measured levels of fractional exhaled nitric oxide (FeNO) serve as a crucial biomarker for detecting airway inflammation [1] The non-type 2 phenotype has been associated with factors such as aging, obesity, and smoking [8] and frequently exhibit a poor response to conventional asthma treatments [6].

Pro-resolving lipid mediators (SPMs) have been recognized for their potential therapeutic effects in asthma management [9]. These bioactive molecules, which include resolvins, protectins, and maresins, are derived from n-3 polyunsaturated fatty acids (n-3 PUFAs) [1] and are associated with attenuating chronic airway inflammation and tissue damage [10] These lipid mediators play a pivotal role in modulating inflammatory responses by regulating the activity of immune cells, such as eosinophils [10], and cytokines (interleukin IL4, IL5, and IL13) with a potential clinical implication for resolving asthma symptoms [1,11]. However, the current literature report conflicting results regarding the efficacy of these mediators in asthma management [12,13].

An in vitro study conducted in 2020 utilizing unrefined salmon oil OmeGO^®^, demonstrated an increased rate of eosinophil apoptosis in eosinophils harvested from allergic individuals. These results underscore the potential anti-allergic properties of OmeGO^®^ and its potential as a therapeutic agent for eosinophil-driven inflammatory conditions [14]. Complementing these findings, a 2023 mouse model study showed a significant reduction in both pulmonary and systemic eosinophilia following OmeGO^®^ administration [15]. Building on previous findings, this prospective, randomized exploratory trial was conducted to evaluate the impact of daily OmeGO^®^ intake over a 20-weeks period in adults with type 2 asthma. Although inflammatory biomarkers provide insight into immune mechanisms, the present investigation emphasizes clinical variables and patient-reported outcomes to provide a multidimensional assessment of clinical asthma control. 

## 2. Results

The demographic and clinical profiles of the OmeGO and placebo groups are detailed in Table 1, with a mean value of eosinophils at 0.34 (10^9^/L) at baseline. The mean heart rate was 68 bpm (OmeGO) vs. 73 bpm (placebo), a difference unlikely to affect outcomes. The participants had a mean age of 57 years (SD 11), with females comprising 57%. In both groups, most participants received a combination therapy of ICS + LABA, while leukotriene receptor antagonist (LTRA) were used by 26% (OmeGO) vs. 17% (placebo). Participant enrollment predominantly occurred between February and August, spanning the late winter to summer seasons. The mean body mass index (BMI) was identical for both groups, recorded at 29 kg/m^2^.

The median time to the first exacerbation (moderate or severe) defined as a composite event (CE) (see Section 4.4) was 37 days (95% CI 9–47) in the OmeGO group and 15 days (95% CI 12–33) in the placebo group. 73% of the participants in the OmeGO experienced at least one exacerbation compared to 82% in the placebo group (*p* = 0.347) (Figure 1).

The weekly rate of composite events (CEs) is illustrated in Figure 2. The frequency of CEs per week was similar in both groups, with an average rate of 0.26 and 0.23, respectively, in the OmeGO and placebo groups during the trial period (*p* = 0.777).

No significant differences in adverse events (AEs) were observed between the OmeGO and placebo groups, with no distinct association in relation to the intervention. One participant from the placebo group was hospitalized during the second week of the trial due to asthma exacerbation, which was deemed unlikely to be related to the trial intervention (*p* = 0.314). Hospitalization due to severe asthma exacerbation was rare in both groups (*p* = 0.314) (Table 2). Importantly, no participants withdrew from the trial due to side effects. Gastrointestinal (GI) symptoms occurred more frequently in the placebo group; however, the differences did not show statistically significant (Appendix A).

Moderate events are presented in Table 3, with the composite outcome scores (COS) totaling 746 (25%) in the OmeGO group and 679 (23%) in the placebo group (*p* = 0.806). Wheezing was the most frequent moderate event (*p* = 0.451). No significant differences were found in coughing, chest tightness, breathlessness, or use of short-acting beta-agonists (SABA). Both groups exhibited an equal 20% reduction in peak expiratory flow (PEF) of the COS calculations (*p* = 0.994). Nighttime awakening requiring rescue medication was infrequent in both groups but occurred more frequently in the OmeGO group (1.7%) compared to the placebo group (0.4%) (*p* = 0.003). See Appendix A for the rate of nighttime awakenings.

In month 5 of the trial, most of the participants in both groups were in the partly controlled status (*p* = 0.382) (Table 4).

At 20 weeks, the ACQ-5-item (ACQ5) scores were declined in both groups, with a greater reduction observed in the placebo group (*p* = 0.211). There were no observed differences between the groups during the trial for FEV_1_ (*p* = 0.661) and FeNO levels (*p* = 0.329). Four participants in the OmeGO group and two in the placebo group required oral corticosteroids (OCS) for severe exacerbations (*p* = 0.392). Compliance with OmeGO capsules was higher in the OmeGO group than in the placebo group (*p* = 0.048) (Table 5).

The mean weekly PEF remained stable and similar across both groups throughout the trial period (*p* = 0.650) (Figure 3).

## 3. Discussion

This exploratory, double-blind, placebo-controlled trial investigated the effects of OmeGO^®^ in adults with type 2 asthma. At least one exacerbation occurred in ≈70% of the OmeGO group and ≈80% of the placebo group, a non-significant difference between the groups. The weekly rate of composite events was comparable across both groups.

Asthma overdiagnosis is a recognized issue, with 30–35% of adults diagnosed not actually having the disease, resulting in a misclassification [16]. In the present trial, 26% of potential participants were excluded at pre-screening due to alternative pulmonary conditions, underscoring the diagnostic challenge. To ensure an accurately diagnosed asthma population, only participants with daily use of inhaled steroids, eosinophils ≥ 0.15, and ACQ5 score ≥ 0.75 were included.

Composite scores are frequently utilized in clinical trials to assess asthma symptomatology and treatment outcomes to enhance powering and signal detection [17]. The incidence of asthma exacerbations defined as composite events did not differ significantly between the groups. However, the OmeGO group had more exacerbation-free days than the placebo group during the trial. Lang et al. [13] defined asthma exacerbation as requiring systemic steroids or urgent medical care, reporting no significant difference in the frequency of events between n-3 PUFA and control groups [13]. In this present trial, moderate exacerbations, evaluated by total COS days and daytime symptom sub-groups, showed no significant differences. In contrast, Barua et al. [18], in a five-arm randomized trial, reported significant improvement in wheezing, breathlessness, cough, and chest tightness (*p* ≤ 0.05) following 3-n PUFA administration compared to placebo [18]. However, insufficient data on the four supplements, particularly eicosapentaenoic acid (EPA)/docosahexaenoic acid (DHA) content, limited direct comparison with other clinical trials. A non-controlled trial in children (4–14 years) reported substantial or partial improvement in 87% of 39 participants in nocturnal cough, nocturnal wheeze, and daytime wheeze following n-3 PUFA treatment. The PEF values (spirometry) indicated improvement post-treatment. The supplement had higher EPA and lower DHA than observed in present trial [19]. Lorensia et al. [20] observed a significant increase in PEF values using a portable flow meter in adults after a one-month trial with the same EPA/DHA ratio as reported by Farjadian et al. [19]. However, 4 out of 27 participants were on daily steroids for asthma control [20], limiting direct comparability to our trial, and no significant differences were found between the groups.

Asthma control is influenced by various external and individual factors, including physical activity, allergen exposure (e.g., pollen), and weather variability [4]. These confounding variables were not accounted for in the present analysis, potentially affecting outcome interpretation. Spring enrollment (60% of participants) may have contributed to symptom prevalence.

Establishing a direct correlation between standardized measurement tools and clinical outcomes in asthma research can be challenging. Murugesan et al. [21] identified a weak correlation between lung function, asthma control, and elevated FeNO in patients on maintenance control treatment. Low FeNO (≥25 ppb) has been associated with disease progression and a higher risk of exacerbations [21], while uncontrolled asthma signifies poor disease control [4]. We report in this trial a mean FeNO at 20 weeks at 64 ppb, with uncontrolled asthma observed in 55–45% of OmeGO participants and 55–30% in the placebo group throughout the trial. Rescue medication (≥4 puffs/day) was rare, suggesting a low frequency of exacerbation. The limited FeNO variation observed in this present trial was consistent with Visser et al. [22], whose systematic review found discrepant FeNO responses to n-3 PUFA supplementation [22].

Sullivan et al. [23] recommend a dual-method approach to asthma evaluation using ACQ5 and FEV_1_% of predicted, but report a weak correlation, particularly in relation to disease severity and asthma control [23]. FEV_1_ values (<60% of predicted) indicates uncontrolled asthma [4,7] and an ACQ5 score ≥ 1.5 signals poor asthma control and increased exacerbation risk [24]. In the present trial, the lowest mean value recorded of FEV_1_ and ACQ5 score showed no significant differences. These findings align with Lang et al. [13] who reported no significant differences in FEV_1_ or asthma control 7- questionnaire (ACQ7) after EPA (3180 mg) and DHA (822 mg) supplementation [13]. Similarly, Farjadian et al. [19] found no significant changes in FEV_1_ following n-3 PUFA supplementation [19], whereas Barua et al. [18] reported significant increase in FEV_1_ volume [18].

The compliance of capsules was high and well tolerated in both groups, with minor GI complaints, which were more frequent in the placebo group. Symptom variability, based on self-assessments, had uncertain clinical significance. Lang et al. [13] noted a non-significant increase in GI symptoms in the n-3 PUFA-treated group, but no causal relation or impact on daily life was established [13].

Adverse events impacting daily life were comparable between the groups. One participant experienced worsening of asthma symptoms after discontinuing maintenance medication, despite clear guidance to continue treatment. This led to temporary deterioration, but symptoms improved quickly upon resuming standard medication.

The present trial possesses several strengths. The trial design ensured comparability between the intervention groups, with random allocation supporting baseline equivalence. Sample size calculations provided sufficient statistical power, while double blinding minimized bias. Both groups received identical capsules, achieving high compliance rates. Validated measures (patient reported outcome measures (PROM), spirometry, FeNO) strengthen internal validity and enable multiple-endpoint evaluation, improving generalizability and reproducibility of findings across broader populations. Incorporating PEF meters and asthma management applications reinforced compliance and ensured accurate composite outcome calculation. Strict adherence to monitoring standards according to Good Clinical Practice safeguarded ethical integrity, data reliability, and trial outcomes. Despite a small sample size, daily reporting over an extended period allowed robust estimation of the weekly CE rate.

The trial has limitations that affect result interpretation. The time to the first exacerbation requires careful analysis, as the exploratory design and a small sample size may have limited statistical power. The trial included 66 participants, fewer than the planned 80, potentially restricting group comparison participants for sufficient statistical power. The eligibility criteria of eosinophil counts ≥0.15 (10^9^/L) may have excluded participants due to day-to-day variability of the eosinophilic count. It is possible that some individuals were misclassified as having asthma, despite inhaled corticosteroids and an ACQ5 score ≥ 0.75. Missing data led to incomplete COS calculations and GINA control metrics calculations. Additionally, the analysis did not account for daily fish consumption, a factor that could have influenced the outcomes.

This trial is the first exploratory clinical trial evaluating the effects of OmeGO^®^ on respiratory symptoms in individuals with type 2 asthma. These non-significant results highlight the need for further research to confirm or refute the observed patterns. We suggest addressing questions such as

Standardize participant selection to include individuals at the same stage of treatment initiation.Evaluate alternative formulations, such as liquids, to enhance tolerability and adherence.Monitor and address GI side effects more proactively during the trial.Consider patient feedback on formulation acceptability during trial design.

## 4. Materials and Methods

This exploratory, prospective, randomized, double-blind, placebo-controlled, multicenter trial was designed to assess the efficacy and safety of OmeGO^®^ in adult patients with type 2 asthma receiving treatment within the GINA steps 2–5 [4]. Participants were randomly assigned to receive either 3000 mg of OmeGO^®^ or placebo, twice daily over a 20-week period. The primary objectives were to evaluate the impact of OmeGO^®^ on a composite outcome score as well as its effect on asthma control.

### 4.1. Recruitment of Participants

Participants were enrolled between June 2022 and July 2024. Key inclusion criteria required a confirmed asthma diagnosis by a pulmonary specialist or general practitioner (GP), age 18–75 years, both sexes, daily treatment with inhaled corticosteroids (ICS) (alone or with maintenance therapy), ACQ5 score ≥ 0.75, and blood eosinophil count ≥ 0.15 10^9^/L. Major exclusion criteria included oral corticosteroids (≤1 month pre-baseline), biological therapies (≤6 months pre-baseline), oral or intravenous antibiotics (≤3 months pre-baseline), fish oil supplements (≤1 months pre-baseline), pregnancy or breastfeeding, and inflammatory bowel disease.

Participant recruitment was conducted through public advertising and pre-screening of medical records. The advertisement campaigns took place between June 2021 and February 2024, and 100 individuals contacted the study team. In parallel, medical records were reviewed over a two-year period spanning August 2021 to September 2023, identifying 718 individuals with documented asthma, of whom 394 were excluded based on eligibility criteria. In total, 324 invitation letters were sent, but 174 declined, and 29 had relocated beyond the trial area. All interested individuals (*n* = 100) and eligible candidates from record screening (*n* = 121) were contacted via telephone for screening. A total of 151 were excluded for not meeting eligibility criteria. Ultimately, 70 participants were enrolled, with 66 completing the trial. A detailed recruitment summary is presented in Figure 4 (CONSORT flow diagram).

Eligible participants were randomized (1:1) using block randomization with variable block sizes (2 and 4). A statistician generated the randomization sequence to ensure allocation blinding. Blinding was maintained throughout the trial, with participants, trial coordinators, and the research unaware of treatment assignments until the data collection was complete and analysis began. The study was conducted across five clinical trial units (CTUs) including Hofseth Biocare Ålesund, Møre og Romsdal Hospital Trust (HMR) hospitals in Ålesund, Molde, Kristiansund, and St. Olavs University Hospital in Trondheim.

### 4.2. Intervention

To ensure blinding integrity, OmeGO^®^ and placebo capsules were identical in size, color, taste, and smell. Each participant received 11 boxes of 90 capsules, instructed to take three capsules (3 × 1000 mg) morning and evening over 20 weeks. OmeGO^®^ contained fresh, unrefined salmon oil produced by Hofseth Biocare ASA and was encapsulated by Pharmatec AS, ensuring batch consistency. Each 1000 mg capsule contained ≥270 mg of polyunsaturated fatty acids (PUFAs), including ≥140 mg of total omega-3 fatty acids and ≥110 mg total omega-6 fatty acid, ≥20 mg eicosapentaenoic acid (EPA); ≥30 mg docosahexaenoic acid (DHA); and ≥10 mg docosapentaenoic acid (DPA) (Appendix A). The total daily intake of PUFAs was 1620 mg, including 840 mg omega-3, 660 mg omega-6, 120 mg EPA, 180 mg DHA, and 60 mg DPA. The total ingredients in the capsules were salmon oil, gelatin, stabilizer (glyzerol), color agent (red iron oxide). The placebo formulation (Pharmatec AS) comprised medium-chain triglycerides (MCTs) derived from caprylic and capric acids, selected for their lack of eosinophilic activity. Except for MCTs, the placebo capsules contained gelatin, stabilizer (glyzerol), and color agent (red iron oxide). At the end of 20 weeks, participants returned the remaining capsules, with compliance assessed via capsule count by the trial coordinators.

### 4.3. Assessment

Participants attended clinical visits at baseline (week 0), week 20, and week 24 at designated CTUs. Baseline and week 20 assessments included vital signs (blood pressure, heart rate, and weight), medical history, blood sampling, and pulmonary function tests. On week 24, only pulmonary function testing was performed. To monitor side effects, documentation of medication changes, and support interventional adherence, structured telephone follow-up calls occurred at weeks 4, 8, 12, and 16 (Appendix A).

Although PEF is generally less reliable than FEV_1_ for assessing lung function [4], its practicality makes it ideal for longitudinal ambulatory monitoring in non-clinical settings [7]. Participants measured PEF by using model 4300 handheld Vitalograph flow meter (EN ISO 23747) (Vitalograph, Ennis, Ireland, year 2020) and performed twice daily assessment for 20 weeks. Results were manually registered in a web-based diary (Zegeba AS (id 814 777 812), Ålesund, Norway). Data was stored on a private cloud server (Software v3.19) with app v1.16 for Android and iOS devices. Participants were instructed to measure PEF within two hours of maintenance asthma medication, preferably at the same time daily. Each session included three consecutive tests, with the highest value recorded. The same PEF meter was used for home and clinical visits. To evaluate weekly measured PEF, a mean value calculated from morning and evening daily measurements and the mean value for every seventh day was captured.

Spirometry was standardized according to the ERS/ATS guideline [25], with at least three repeated tests to ensure consistency, the software program selected the most accomplished test for recording. Four CTUs used model SPS330 handheld Spirare sensor with v3.42.1.2933 Spirare software (Diagnostica AS, Oslo, Norway), and model 2050-1 disposable NDD Spirette mouthpiece (ndd Medizintechnik AG, Zürich, Switzerland). One CTU applied handheld device Vyntus APS^®^ (Vyaire Medical, Mettmann, Germany) with v3.10 Sentry-Suite^®^ software (CareFusion, Hoechberg, Germany) and V-892381 MicroGard IIB^®^ disposable mouthpiece (Vyaire Medical, Hoehberg, Germany). FeNO assessment followed the ATS guideline [21], using model F 09G 100 168 Vivatmo Pro Base station and F 09G 100 078 handheld unit with F 09G 100 227disposable mouthpiece Vivatmo oxycap (Bosch Healthcare Solutions, Waiblingen, Germany) in four CTUs. One CTU used FeNO device 12-1000 NIOX VERO^®^ with 12-1810 NIOX VERO^®^ disposable mouthpiece (Aerocrine AB, Solna, Sweden). The mean value of two FeNO measurements was calculated. Participants were instructed to abstain from drinking, chewing gum, or tobacco for at least one hour prior to spirometry and FeNO, with SABA use restricted to six hours pre-assessment. All pulmonary function assessments were conducted by licensed physiotherapists and nurses, trained in standardized equipment use, ensuring measurement accuracy and reliability.

### 4.4. Primary Outcome: The Occurrence of Exacerbations

The primary endpoint was the occurrence of moderate or severe asthma exacerbations, comparing the OmeGO and placebo groups. Moderate asthma symptoms were monitored twice-daily dairy recordings, alongside severe clinical event documentation. These parameters formed a composite outcome, referred to as composite events (CEs), serving as the primary indicator of treatment efficacy.

Symptom diaries are widely recognized for monitoring clinical deterioration in asthma control in clinical trials [26]. To enhance exacerbation detection, the composite endpoint (CompEx, Wrexham, UK), integrates daily symptom fluctuations with sustained worsening beyond a predefined baseline threshold, primarily for assessing the effectiveness of medical therapies in exacerbation management [8,17]. CompEx has been extensively utilized in asthma clinical trials, improving exacerbation event identification [27,28]. In the present trial, symptom diaries (Zegeba AS) were based on the CompEx model [17] and GINA guidelines [7] to systematically monitor asthma control throughout the intervention period.

The CE was defined as the occurrence of either moderate or severe asthma exacerbation. Severe events were identified via participant self-reports, including oral corticosteroids (OCS) use or hospitalization due to asthma exacerbation. Moderate exacerbation was integrated into the composite endpoint score (COS), serving as an indicator of worsening symptoms from baseline. Moderate asthma symptoms criteria included nighttime awakenings requiring rescue medication, increased rescue medication use (≥4 inhalations/day), ≥20% PEF reduction, and worsening daily symptoms (Appendix A). Daily symptoms, assessed via PROM, included coughing, wheezing, chest tightness, and breathlessness (Appendix A), with at least one point in two distinct questions. A COS was recorded when moderate asthma symptoms worsened for two consecutive days relative to baseline.

The maximum number of COS recorded per day was one. Sub-scores represented individual COS variables meeting the two-day criterion, resulting in total of sub-scores exceeding the total complete COS days. The theoretical maximum number of per change COS calculations was 4653 for each group (141 days of follow-up, 33 participants per group), though actual values were lower due to incomplete diaries, which led to missing composite variable data.

### 4.5. Secondary Outcome: Asthma Control and Safety Profile

Asthma control was assessed using validated tools, following established clinical guidelines [4,7]. Data derived from PROMs (Appendix A) were applied to determine asthma control status in accordance with the GINA classification system (Appendix A). The GINA classification system evaluates control based on daytime symptom frequency and use of a SABA-reliever more than twice per week, nighttime awakenings, and activity limitations due to asthma. According to GINA criteria, asthma is well-controlled if none of these indicators occurs over four weeks, while 1–2 episodes within the same timeframe classify it as partly controlled [4].

The ACQ5 was administered every four weeks via participant diaries. This validated, five-item, self-reported tool assesses asthma control in clinical trials, evaluating nighttime awakenings, morning symptoms, activity limitations, shortness of breath, and wheezing over the past week. Scores range from 0 (completely controlled) to 6 (severely uncontrolled), with higher scores indicating poorer asthma management [23,29,30]. Permission to use the Norwegian version of the ACQ5 was granted by Dr. Elizabeth Juniper in April 2021.

GI symptoms and events were systematically recorded during clinical and telephone follow-up visits using a standardized questionnaire. Documented symptoms included regurgitation, diarrhea, nausea, abdominal pain, constipation, and general GI discomfort. Mild untoward GI symptoms were tolerable and non-disruptive to daily activities, while adverse events (AEs) were moderate symptoms causing affecting daily functioning. Serious adverse events (SAEs) required hospitalization for at least 24 h. The causal relationship between the intervention and reported GI symptoms/events was self-assessed and categorized into five grades: unrelated, unlikely, possible, probable, or definite. Follow-up procedures were adjusted based on symptoms severity to ensure proper participant management.

### 4.6. Sample Size and Power Calculation

This exploratory study was designed to detect a decrease in the rate of exacerbations of 40% (i.e., a rate ratio of 0.6), assuming a significance level of 5%, a power of 80% and an individual event rate of 0.01 pr day in the control group [31]. To achieve this a sample size of 80, 40 in each group was required.

### 4.7. Statistical Analysis

The principal investigator and the statistician signed a statistical analysis plan (SAP) prior to unblinding the captured data, and the statistical analysis was initiated in September 2024. Means and standard deviations (SD) described continuous variables; independent samples *t*-tests tested the differences between means. Absolute numbers and percentages described the categorical variables; Chi-square tests tested the differences in the distributions. A Kaplan–Meier estimator analyzed the time to the first composite event, and the log-rank test compared the differences between the groups. Generalized linear models with a log-link function estimated relative risks and calculated robust standard errors clustered at each subject. The rate of weekly composite events was the total number of composite events divided by the total number of person-days for each week, using information from individuals with non-missing data only. We assumed the Poisson distribution in the calculation of confidence intervals (95% CI). The significance level was set to 5%. All analyses were performed using StataNow 18.5.

### 4.8. Informed Consent and Registration

All participants were provided with both oral and written information detailing the purpose, procedures, and potential risks associated with the trial. Prior to enrollment, written informed consent was obtained from participants to ensure ethical compliance and voluntary participation. The trial was registered at ClinicalTrials.gov (Identifier: NCT05137132) and at the Norwegian clinical trials platform, HelseNorge.no, in accordance with national regulatory requirements.

## 5. Conclusions

This randomized exploratory trial found no significant differences in asthma exacerbation rates between OmeGO^®^ and the placebo as shown by the respiratory symptoms, asthma control, and pulmonary function results. Although not statistically significance, the difference in median time to first exacerbation and the number of patients with no exacerbations warrant further investigation.

## Figures and Tables

**Figure 1 marinedrugs-23-00328-f001:**
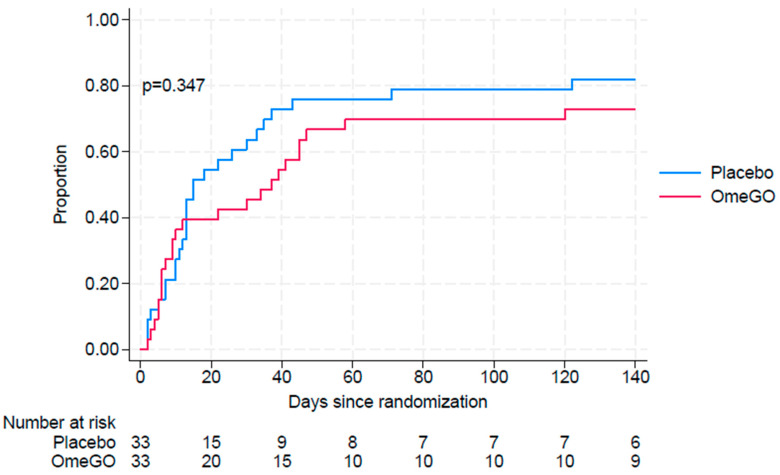
Time to first composite event. Time to first composite event (CE) and includes both moderate and severe events.

**Figure 2 marinedrugs-23-00328-f002:**
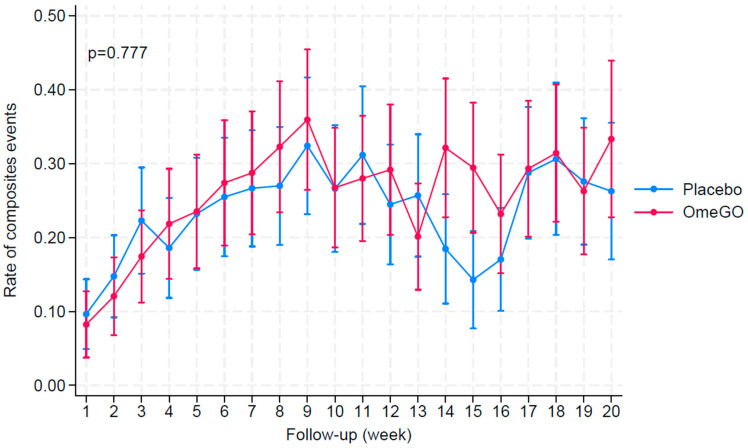
The rate of days with composite events (CEs) per week. The rate for weekly CEs includes both moderate and severe events. This rate is calculated as the cumulative number of composite events divided by the total number of person-days for each week.

**Figure 3 marinedrugs-23-00328-f003:**
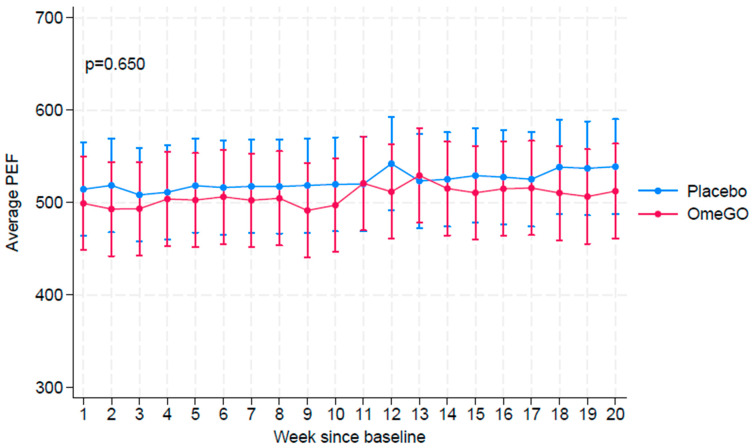
Weekly average peak expiratory flow (PEF). Confidence intervals (95% CI) are calculated assuming normal distribution. *p*-value from testing the null hypothesis of the overall difference in means being equal to zero (0).

**Figure 4 marinedrugs-23-00328-f004:**
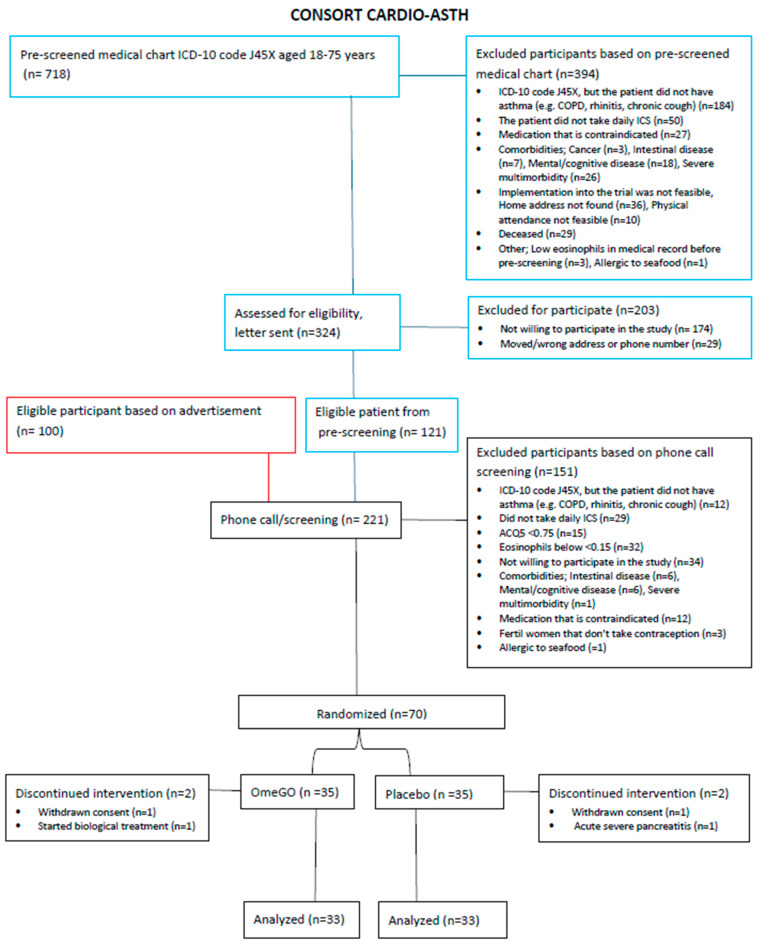
Consort flow diagram illustrating the flow of participant recruited, allocated, and total included.

**Table 1 marinedrugs-23-00328-t001:** Baseline characteristics of the participants.

	OmeGO (N = 35)	Placebo (N = 35)
	Screening	
Eosinophils in blood (10^9^/L), mean (SD)	0.44 (0.44)	0.36 (0.30)
ACQ5 ^1^, mean (SD)	1.9 (0.8)	1.9 (0.9)
	Baseline	
Demographic:		
Age, mean (SD)	58 (11)	55 (11)
Female, n (%)	21 (60)	19 (54)
Body mass index (kg/m^2^), mean (SD)	29 (6)	29 (5)
Inclusion month, n (%)		
February–August	20 (57)	22 (63)
September–January	15 (43)	13 (37)
Clinical characteristics:		
Systolic blood pressure (mm Hg), mean (SD)	135 (18)	131 (12)
Diastolic blood pressure (mm Hg), mean (SD)	84 (9)	82 (7)
Heart rate (beats/minute), mean (SD)	68 (9)	73 (13)
Cardiovascular disease ^2^, n (%)	10 (29)	8 (23)
Eosinophils in blood (10^9^/L), mean (SD)	0.36 (0.27)	* 0.32 (0.27)
Pulmonary assessment:		
FEV_1_ in % of predicted value ^3^, mean (SD)	87 (17)	82 (15)
FEV_1_ (L/s) Below LLN ^3^, n (%)	9 (26)	13 (37)
FEV_1_ (L/s) Above LLN ^3^, n (%)	26 (74)	22 (63)
FeNO (ppb) ^4^, mean (SD)	57 (42)	68 (65)
Asthma therapy ^5^:		
ICS, n (%)	2 (6)	4 (11)
ICS + LABA, n (%)	27 (77)	28 (80)
ICS + LAMA, n (%)	1 (3)	0 (0)
ICS + LABA + LAMA, n (%)	5 (14)	3 (9)
LTRA, n (%)		
Yes	9 (26)	6 (17)
No	26 (74)	29 (83)

^1^ ACQ5 = Asthma Control Questionnaire, 5 questions- ^2^ Heart failure, hypertension, ischemic, and atrial fibrillation. ^3^ Forced expiratory volume (FEV_1_) the amount (liter) of air forced out in 1 s; below and above lower limit of normal (LLN). ^4^ Fractional exhaled nitric oxide (FeNO) (ppb). ^5^ Asthma inhalation therapy: ICS = corticosteroid; LAMA = long-acting muscarinic antagonist; LABA = long-acting β2-agonist; SABA= short-acting β2-agonist; LTRA = leukotriene receptor antagonist. * N = 34 due to one participant withdrawing from study the day after randomization. *t*-test for comparison of means; Chi-square test for comparison of n (%).

**Table 2 marinedrugs-23-00328-t002:** Safety monitoring.

Events	OmeGO (N = 33) **	Placebo (N = 33) **	*p*-Value *
Adverse Event (AE)			
Post-traumatic stress disorder (PTSD)		1 ^(a)^	0.314
Increased waist measurement/gained weight	1 ^(b)^		0.314
Change in pre-existing medical conditions, increased asthma symptoms	1 ^(b)^		0.314
Nausea		1 ^(c)^	0.314
Dyspepsia (obstipation and bloating)	1 ^(b)^		0.314
Stomach pain		2 ^(c)^	0.151
Diarrhea	2 ^(a)^	1 ^(a)^	0.555
Serious Adverse Event (SAE) ***			
Urine tract infection		1 ^(a)^	0.314
Asthma exacerbation		1 ^(b)^	0.314
Complication post cardio atrial fibrillation		1 ^(a)^	0.314
Idiopathic pancreatitis		1 ^(b)^	0.314

Reported adverse events (AEs) and serious adverse events (SAEs). * Chi-square test for differences in proportion. ** Causal relation to intervention. ^(a)^ Unrelated. ^(b)^ Unlikely. ^(c)^ Possible. Participants who reported events were assigned to the corresponding row based on the specific event category. *** Hospitalization ≥ 24 h.

**Table 3 marinedrugs-23-00328-t003:** Composite outcome score (COS) and sub-scores of moderate events.

	OmeGO (N = 33)		Placebo (N = 33)			
Number of days with two consecutive days of potential COS calculation	Days * (2965)		Days * (2920)			
	Number of events (%)	N ***	Number of events (%)	N ***	RR (95% CI)	*p*-Value **
Total days with COS ****	746 (25)	24	679 (23)	27	1.08 (0.58–2.03)	0.806
	Sub-scores					
Days with daytime symptoms and use of SABA	488 (17)		486 (17)		0.99 (0.46–2.12)	0.977
Days with coughing	272 (9)		310 (11)		0.86 (0.34–2.18)	0.757
Days with wheezing	688 (23)		496 (17)		1.37 (0.61–3.07)	0.451
Days with chest tightness	334 (11)		359 (12)		0.92 (0.33–2.57)	0.868
Days with breathlessness	363 (12)		457 (16)		0.78 (0.29–2.09)	0.624
Days with use of SABA	219 (7)		326 (11)		0.66 (0.32–1.39)	0.274
Nighttime awakening with use of rescue medication	49 (1.7)		5 (0.2)		9.65 (2.11–44.17)	0.003
Increased dose rate of ≥4 puffs/day of rescue medication	11 (0.4)		12 (0.4)		0.90 (0.11–7.53)	0.925
Number of events with ≥20% reduction in peak expiratory flow (PEF)	284 (10)		281 (10)		1.00 (0.28–3.57)	0.994

* The total of maximum number of days where a COS perchance could be measured is 4653. Due to incomplete registration, the actual number of days where a COS could be measured is less (2965 and 2920). ** *p*-value from testing the null (0) hypothesis of the risk ratio being equal to one (1). *** Number of participants with at least one registration of COS. **** The total number of days represents the overall total of COS for a set period.

**Table 4 marinedrugs-23-00328-t004:** Global Initiative for Asthma (GINA) asthma control.

	OmeGO (N = 33)			Placebo (N = 33)			
	Well Controlled	Partly Controlled	Un Controlled	Well Controlled	Partly Controlled	Un Controlled	
	N (%)	N (%)	N (%)	N (%)	N (%)	N (%)	*p*-Value *
Month 1	3 (9)	12 (36)	18 (55)	0 (0)	15 (45)	18 (55)	0.189
Month 2	3 (9)	11 (33)	19 (58)	1 (3)	17 (52)	15 (45)	0.252
Month 3	3 (9)	12 (36)	18 (55)	5 (15)	14 (42)	14 (43)	0.562
Month 4	3 (9)	17 (52)	13 (39)	4 (12)	15 (46)	14 (42)	0.859
Month 5	2 (6)	16 (49)	15 (45)	4 (12)	19 (58)	10 (30)	0.382

Table 4 shows the number of levels of wellcontrolled, partly controlled, and uncontrolled per study group according to Global Initiative for Asthma (GINA). * Chi-square test for differences in proportion.

**Table 5 marinedrugs-23-00328-t005:** Secondary outcomes.

	OmeGO (N = 33) (95% CI)	Placebo (N = 33) (95% CI)	*p*-Value *
ACQ5-score ^1^ mean			
Baseline	1.6 (1.3–1.9)	1.4 (1.2–1.7)	
20 weeks	1.5 (1.2–1.9)	1.1 (0.8–1.3)	0.211
FEV_1_% of predicted value ^2^, mean			
Baseline	87 (81–93)	82 (76–87)	
20 weeks	87 (81–93)	82 (76–88)	0.661
FeNO ppb ^3^, mean			
Baseline	57 (42–72)	69 (46–93)	
20 weeks	61 (49–74)	66 (45–87	0.329
Concomitant medication ^4^, n (%)			
Increased	5 (15.2%)	4 (12.1%)	
Reduced	3 (9.1%)	5 (15.2%)	0.729
Oral corticosteroids (OCS) ^5^, n (%)	4 (12%)	2 (6%)	0.392
Compliance of intervention capsules taken ^6^, mean	96 (94–98)	92 (88–96)	0.048

^1^ ACQ5 = Asthma Control Questionnaire, 5 questions. Mean change in score. ^2^ Forced expiratory volume (FEV_1_) of the amount of air forced out in 1 s measured by spirometry. Percentage of predicted value calculated. ^3^ Fractional exhaled nitric oxide (FeNO), level of nitric oxide in part per billion (ppb). Mean change in ppb. ^4^ Change in concomitant asthma standard medication. The adjustment is calculated from standard medication at baseline visit. ^5^ Number of participants with use of systemic oral corticosteroids (OCS) tablets. ^6^ Percentage of total capsules administered. * *t*-test for comparison of means; Chi-square test n (%).

## Data Availability

The data generated and analyzed in this trial are stored securely in accordance with the General Data Protection Regulation (GDPR) and the requirements set forth by REK Central. Due to ethical and legal restrictions, the full dataset is not available. However, data may be made available on reasonable request to the corresponding author dag.hoff@helse-mr.no (D.A.L.H.) and cc@hofsethbiocare.no (C.C.).

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
