# Peer review of "The Efficacy of Daily Salmon Oil for Adult Type 2 Asthma: An Exploratory Randomized Double-Blind Trial"

_marinedrugs, 2025, doi:10.3390/md23080328_

Round 1

Reviewer 1 Report

Comments and Suggestions for Authors

I have reviewed the manuscript submitted by Molsaeter and colleagues. This is an interesting study addressing the effect of a specific type of fish oil on asthma exacerbations in a Norwegian adult population. While the study does not appear to show significant benefit on primary outcomes, this work merits to be published as further optimization of the treatment protocol may reveal specific methodological aspects that need to be addressed to achieve future success with a complicated matrix such as fish oil. I have the following major and minor comments the authors may want to address.

Major comments:

- This study employed an encapsulated salmon oil as the active principle. A full description of the following parameters is necessary to properly this edible oil, as is customary by ingredient producers, and to ascertain that the quality of the oil meet global and regional regulations and standards: Full fatty acid profile, oxidative quality of the encapsulated oil (Peroxide Value, Anisidine Value and Totox number), measured levels of the following environmental contaminants in the oil: dioxins & furans, dioxin-like PCBs, PCBs, heavy metals (lead, cadmium, mercury and inorganic arsenic), . The oil supplier can provide the measured values for these parameters. Optional contaminants to include for which European regulations are in development: phthalates, bisphenol A and brominated flame retardants. The information can be summarized in a table in the Supplementary Information

- Whereas most fish oils commercialized for human consumption are refinedin order to make fish oils suitable for human consumption, the OmeGo salmon oil used for this study is unrefined. Please discuss the reason why an unrefined oil was employed. In addition - while it is certainly possible to produce an unrefined fish oil that meets international quality requirements, the lack of refining warrants a full disclosure of the levels of environmental contaminants, as mentioned in the previous comment.

- Salmon oils in general contain modest levels of EPA and DHA, compared to some other fish oils and concentrates that are more commonly used in food supplements or pharmaceutical APIs. While the manuscript highlights the importance of EPA and DHA for the hypothesized benefit for asthma symptoms, this study could not establish a statistically significant benefit for the use of OmeGo salmon oil at the dose and exposure time that was evaluated. It is however not possible to confirm that the dose used was sufficient to raise tissue levels of omega-3 LCPUFA with the current dosing regimen. In order to do that, a baseline measurement of the Omega-3 Index, or similar readout, should have been taken, and followed during the course of the study, to demonstrate that there was significant increase in red blood cell membrane levels. Why was this not done, if there is much recognition in the literature that this is an essential check to make for successful delivery of EPA/DHA supplementation? In any case, the monitoring of the Omega-3 Index (or similar readout) should be included in the suggestions for future work on OmegGo oil supplementation for asthma therapy (starting on line 264).

- Lines 69-76 about the role of SPMs can be removed. This study did not address the role of SPMs in asthma at all, and it seems superfluous. If anything, a few words can be dedicated to SPMs in the Discussion section. It is also odd to see that SPMs are mentioned so prominently whereas the inhibition of arachidonic acid-derived mediators is a far more established option in asthma therapy (leukotriene receptor antagonists, FLAP inhibition) than novel approaches using inflammation resolving derivatives of EPA or DHA. In this regard, the request to include a full fatty acid profile of the OmegoGo oil (see comment above) is of relevance, especially with regard to declaring the content of arachidonic acid, as precursor of both pro- and anti-inflammatory eicosanoids active in asthma, (and linoleic acid), alongside of EPA and DHA (mentioned in lines 312-315).

Minor comments:

- Up to 26% of study participants were reported to receive leukotriene receptor antagonists. Yet no mention of this drug class is provided in the introduction section line 50-58. Why?

- Line 104 – can refer to section 4.4 for an explanation of “composite event”, which is abbreviated here but unclear what it means

- Line 313 – The authors report a content of 270 mg of polyunsaturated fatty acid per 1000 mg capsule. But in brackets it is stated that this is all “n-3 PUFAs”. This is surprising as fish oils will contain n-6 PUFA as well. EPA, DHA and DPA reportedly amount to 60 mg of PUFA, while 80 mg of PUFA is unaccounted for. That means more than half of the total weight of PUFA are other n-3 PUFA, or perhaps n-6 PUFA species. The authors can make a greater effort to provide a better description of the composition here. The type of DPA is DPA n-3 or DPA n-6, or both are reported together? Also, if the weight of the capsule is exactly 1000 mg, what is the weight of the encapsulated oil?

- Section 4.2 – for the sake of completeness, they type of capsule material should be reported (i.e. type of gelatin, if used), as well as the antioxidant composition employed (or if stability is preserved by the composition of naturally-present antioxidants, which have not been removed by refining, their types and levels).

- Line 367 – Please provide a short description of what types of events or exacerbation make up the composite event (CE) that was quantified.

Typos:

  • Line 79 – harvested
  • Table 1 – August
  • Table 2 – not all rows are fully aligned (values not clearly assigned to row above or below)

Reviewer 2 Report

Comments and Suggestions for Authors

This manuscript reports findings from an RCT of omega-3 PUFAs in patients with type-2 asthma. Intervention period was 20 weeks. Outcomes were clinically related. Overall there was no effect although time to first composite event was longer in the omega-3 group, although the p value for that is not given. The study recruited fewer patients than intended (70 with 66 completers compared with 80 intended) and the effect size was smaller than used in the power calculation, suggesting the study is underpowered. 

This is a study of bioactive omega-3s, yet nowhere is it stated what the daily intake of EPA+DHA is. This should be included in the abstract and explictly stated in the methods.

There is a statement in the abstract and again elsewhere that the omega-3 preparation contains SPMs, yet nowhere are we told which SPMs and how much. If they are important to the study this information needs to be provided. Otherwise remove mention of SPMs. 

The study would be strengthened by knowing the EPA and DHA status at study entry and at week 20 in the two groups. Not having that information is a limitation and must be mentioned. Otherwise analyse your blood samples if you have them.

Specific comments:

  1. Line 28. Specify amount of EPA and DHA per day.
  2. Line 28. Which SPMs? How much?
  3. Line 32. p value?
  4. Line 79. harvest -> harvested
  5. Line 80. the anti-allergic -> the potential anti-allergic
  6. Line 89. 0,34 -> 0.34
  7. Line 94. Year?
  8. Line 105. p value?
  9. Line 146. Delete "are"
  10. Line 224-225. I do not understand what you mean here.
  11. Line 290. September 2021?
  12. Line 213-314. Unclear. Is n-3 PUFAs the abbreviation for polyunsaturated fatty acids, which is what you say? If so how can 270 mg n-4 PUFA and >= 140 total omega-3 fatty acids be compatible. Please be clear about EPA, DPA and DHA dosage. 
  13. Previously you mentioned SPMs as being important in the intervention but section 4.2 makes no mention of them.
  14. Line 330. Ireland
  15. Line 330. performing -> performed
  16. Line 412-413. How many patients are required to achieve this. Power calculation should explicitly state the sample size. 
  17. Line 413. How was the effect size of 40% reduction arrived at? 

Round 2

Reviewer 2 Report

Comments and Suggestions for Authors

This is a good revision that addresses most of the points I raised previously. There are a small number of things that require attention:

  1. Line 27. You do not need the insertion "the salmon oil"
  2. Line 27. You still say "enriched with pre-resolving mediators". This should be deleted.
  3. Line 324. Delete "approximately"
  4. Line 325. PUFA -> PUFAs
  5. Line 325. Delete "(n-3 PUFAs)"
  6. Line 326. Delete "also contained omega-3 ("